# Hamming Ball Auxiliary Sampling for Factorial Hidden Markov Models

**Michalis K. Titsias**
Department of Informatics
Athens University of Economics and Business
mtitsias@aueb.gr

**Christopher Yau**
Wellcome Trust Centre for Human Genetics
University of Oxford
cyau@well.ox.ac.uk

## Abstract

We introduce a novel sampling algorithm for Markov chain Monte Carlo-based Bayesian inference for factorial hidden Markov models. This algorithm is based on an auxiliary variable construction that restricts the model space allowing iterative exploration in polynomial time. The sampling approach overcomes limitations with common conditional Gibbs samplers that use asymmetric updates and become easily trapped in local modes. Instead, our method uses symmetric moves that allows joint updating of the latent sequences and improves mixing. We illustrate the application of the approach with simulated and a real data example.

## 1 Introduction

The *hidden Markov model* (HMM) [1] is one of the most widely and successfully applied statistical models for the description of discrete time series data. Much of its success lies in the availability of efficient computational algorithms that allows the calculation of key quantities necessary for statistical inference [1, 2]. Importantly, the complexity of these algorithms is linear in the length of the sequence and quadratic in the number of states which allows HMMs to be used in applications that involve long data sequences and reasonably large state spaces with modern computational hardware. In particular, the HMM has seen considerable use in areas such as bioinformatics and computational biology where non-trivially sized datasets are commonplace [3, 4, 5].

The *factorial hidden Markov model* (FHMM) [6] is an extension of the HMM where multiple independent hidden chains run in parallel and cooperatively generate the observed data. In a typical setting, we have an observed sequence $Y = (\mathbf{y}_1, \ldots, \mathbf{y}_N)$ of length $N$ which is generated through $K$ binary hidden sequences represented by a $K \times N$ binary matrix $X = (\mathbf{x}_1, \ldots, \mathbf{x}_N)$. The interpretation of the latter binary matrix is that each row encodes for the presence or absence of a single feature across the observed sequence while each column $\mathbf{x}_i$ represents the different features that are active when generating the observation $\mathbf{y}_i$. Different rows of $X$ correspond to independent Markov chains following

$$p(x_{k,i}|x_{k,i-1}) = \begin{cases} 1 - \rho_k, & x_{k,i} = x_{k,i-1}, \\ \rho_k, & x_{k,i} \neq x_{k,i-1}, \end{cases} \qquad (1)$$

and where the initial state $x_{k,1}$ is drawn from a Bernoulli distribution with parameter $\nu_k$. All hidden chains are parametrized by $2K$ parameters denoted by the vectors $\boldsymbol{\rho} = \{\rho_k\}_{k=1}^K$ and $\mathbf{v} = \{v_k\}_{k=1}^K$. Furthermore, each data point $\mathbf{y}_i$ is generated conditional on $\mathbf{x}_i$ through a likelihood model $p(\mathbf{y}_i|\mathbf{x}_i)$ parametrized by $\boldsymbol{\phi}$. The whole set of model parameters consists of the vector $\boldsymbol{\theta} = (\boldsymbol{\phi}, \boldsymbol{\rho}, \mathbf{v})$ which determines the joint probability density over $(Y, X)$, although for notational simplicity we omit reference to it in our expressions. The joint probability density over $(Y, X)$ is written in the form

$$p(Y, X) = p(Y|X)p(X) = \left( \prod_{i=1}^N p(\mathbf{y}_i|\mathbf{x}_i) \right) \left( \prod_{k=1}^K p(x_{k,1}) \prod_{i=2}^N p(x_{k,i}|x_{k,i-1}) \right), \qquad (2)$$

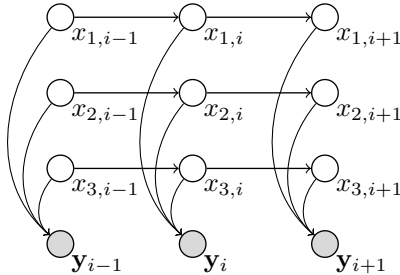

Figure 1: Graphical model for a factorial HMM with three hidden chains and three consecutive data points.

and it is depicted as a directed graphical model in Figure 1.

While the HMM has enjoyed widespread application, the utility of the FHMM has been relatively less abundant. One considerable challenge in the adoption of FHMMs concerns the computation of the posterior distribution $p(X|Y)$ (conditional on observed data and model parameters) which comprises a fully dependent distribution in the space of the $2^{KN}$ possible configurations of the binary matrix $X$. Exact Monte Carlo inference can be achieved by applying the standard forward-filtering-backward-sampling (FF-BS) algorithm to simulate a sample from $p(X|Y)$ in $O(2^{2K}N)$ time (the independence of the Markov chains can be exploited to reduce this complexity to $O(2^{K+1}KN)$ [6]). Joint updating of $X$ is highly desirable in time series analysis since alternative strategies involving conditional single-site, single-row or block updates can be notoriously slow due to strong coupling between successive time steps. However, although the use of FF-BS is quite feasible for even very large HMMs, it is only practical for small values of $K$ and $N$ in FHMMs. As a consequence, inference in FHMMs has become somewhat synonymous with approximate methods such as variational inference [6, 7].

The main burden of the FF-BS algorithm is the requirement to sum over all possible configurations of the binary matrix $X$ during the forward filtering phase. The central idea in this work is to avoid this computationally expensive step by applying a restricted sampling procedure with polynomial time complexity that, when applied iteratively, gives exact samples from the true posterior distribution. Whilst regular conditional sampling procedures use *locally asymmetric* moves that only allow one part of $X$ to be altered at a time, our sampling method employs *locally symmetric* moves that allow localized joint updating of all the constituent chains making it less prone to becoming trapped in local modes. The sampling strategy adopts the use of an auxiliary variable construction, similar to slice sampling [8] and the Swendsen-Wang algorithm [9], that allows the automatic selection of the sequence of restricted configuration spaces. The size of these restricted configuration spaces is user-defined allowing control over balance between the sampling efficiency and computational complexity. Our sampler generalizes the standard FF-BS algorithm which is a special case.

## 2 Standard Monte Carlo inference for the FHMM

Before discussing the details of our new sampler, we first describe the limitations of standard conditional sampling procedures for the FHMM. The most sophisticated conditional sampling schemes are based on alternating between sampling one chain (or a small block of chains) at a time using the FF-BS recursion. However, as discussed in the following and illustrated experimentally in Section 4, these algorithms can easily become trapped in local modes leading to inefficient exploration of the posterior distribution.

One standard Gibbs sampling algorithm for the FHMM is based on simulating from the posterior conditional distribution over a single row of $X$ given the remaining rows. Each such step can be carried out in $O(4N)$ time using the FF-BS recursion, while a full sweep over all $K$ rows requires $O(4KN)$ time. A straightforward generalization of the above is to apply a block Gibbs sampling where at each step a small subset of chains is jointly sampled. For instance, when we consider pairs of chains the time complexity for sampling a pair is $O(16N)$ while a full sweep over all possible pairs requires time $O(16\frac{K(K-1)}{2}N)$.

$$\begin{pmatrix} .. & 1 & 0 & .. \\ .. & 0 & 0 & .. \\ .. & 1 & 0 & .. \end{pmatrix} \nRightarrow \begin{pmatrix} .. & 0 & 1 & .. \\ .. & 0 & 1 & .. \\ .. & 0 & 0 & .. \end{pmatrix} \qquad \begin{pmatrix} .. & 1 & 0 & .. \\ .. & 0 & 0 & .. \\ .. & 1 & 0 & .. \end{pmatrix} \Rightarrow \begin{pmatrix} .. & 1 & 0 & .. \\ .. & 0 & 1 & .. \\ .. & 0 & 0 & .. \end{pmatrix} \Rightarrow \begin{pmatrix} .. & 0 & 1 & .. \\ .. & 0 & 1 & .. \\ .. & 0 & 0 & .. \end{pmatrix}$$

$$\qquad X^{(t)} \qquad\qquad X^{(t+1)} \qquad\qquad X^{(t)} \qquad\qquad\qquad U \qquad\qquad\qquad X^{(t+1)}$$

$$\text{(a)} \qquad\qquad\qquad\qquad\qquad\qquad \text{(b)}$$

Figure 2: Panel (a) shows an example where from a current state $X^{(t)}$ it is impossible to jump to a new state $X^{(t+1)}$ in a single step using block Gibbs sampling on pairs of rows. In contrast, Hamming ball sampling applied with the smallest valid radius, i.e. $m = 1$, can accomplish such move through the intermediate simulation of $U$ as illustrated in (b). Specifically, simulating $U$ from the uniform $p(U|X)$ results in a state having one bit flipped per column compared to $X^{(t)}$. Then sampling $X^{(t+1)}$ given $U$ flips further two bits so in total $X^{(t+1)}$ differs by $X^{(t)}$ in four bits that exist in three different rows and two columns.

While these schemes can propose large changes to $X$ and be efficiently implemented using forward-backward recursions, they can still easily get trapped to local modes of the posterior distribution. For instance, suppose we sample pairs of rows and we encounter a situation where, in order to escape from a local mode, four bits in two different columns (two bits from each column) must be jointly flipped. Given that these four bits belong to more than two rows, the above Gibbs sampler will fail to move out from the local mode no matter which row-pair, from the $\frac{K(K-1)}{2}$ possible ones, is jointly simulated. An illustrative example of this phenomenon is given in Figure 2(a).

We could describe the conditional sampling updates of block Gibbs samplers as being *locally asymmetric*, in the sense that, in each step, one part of $X$ is restricted to remain unchanged while the other part is free to change. As the above example indicates, these locally asymmetric updates can cause the chain to become trapped in local modes which can result in slow mixing. This can be particularly problematic in FHMMs where the observations are jointly dependent on the underlying hidden states which induces a coupling between rows of $X$. Of course, locality in any possible MCMC scheme for FHMMs seems unavoidable, certainly however, such a locality does not need to be asymmetric. In the next section, we develop a *symmetrically local* sampling approach so that each step gives a chance to any element of $X$ to be flipped in any single update.

## 3 Hamming ball auxiliary sampling

Here we develop the theory of the *Hamming ball* sampler. Section 3.1 presents the main idea while Section 3.2 discusses several extensions.

### 3.1 The basic Hamming ball algorithm

Recall the $K$-dimensional binary vector $\mathbf{x}_i$ (the $i$-th column of $X$) that defines the hidden state at $i$-th location. We consider the set of all $K$-dimensional binary vectors $\mathbf{u}_i$ that lie within a certain Hamming distance from $\mathbf{x}_i$ so that each $\mathbf{u}_i$ is such that

$$\mathrm{h}(\mathbf{u}_i, \mathbf{x}_i) \leq m. \tag{3}$$

where $m \leq K$. Here, $\mathrm{h}(\mathbf{u}_i, \mathbf{x}_i) = \sum_{k=1}^{K} I(u_{k,i} \neq x_{k,i})$ is the Hamming distance between two binary vectors and $I(\cdot)$ denotes the indicator function. Notice that the Hamming distance is simply the number of elements the two binary vectors disagree. We refer to the set of all $\mathbf{u}_i$s satisfying (3) as the $i$-th location *Hamming ball* of radius $m$. For instance, when $m = 1$, the above set includes all $\mathbf{u}_i$ vectors restricted to be the same as $\mathbf{x}_i$ but with at most one bit flipped, when $m = 2$ these vectors can have at most two bits flipped and so on. For a given $m$, the cardinality of the $i$-th location Hamming ball is

$$M = \sum_{j=0}^{m} \binom{K}{j}. \tag{4}$$

For $m = 1$ this number is equal to $K + 1$, for $m = 2$ is equal to $\frac{K(K-1)}{2} + K + 1$ and so on. Clearly, when $m = K$ there is no restriction on the values of $\mathbf{u}_i$ and the above number takes its maximum value, i.e. $M = 2^K$. Subsequently, given a certain $X$ we define the full path Hamming

ball or simply Hamming ball as the set

$$\mathcal{B}_m(X) = \{U; \mathrm{h}(\mathbf{u}_i, \mathbf{x}_i) \leq m, i = 1, \ldots, N\}, \tag{5}$$

where $U$ is a $K \times N$ binary matrix such that $U = (\mathbf{u}_1, \ldots, \mathbf{u}_N)$. This Hamming ball, centered at $X$, is simply the intersection of all $i$-th location Hamming balls of radius $m$. Clearly, the Hamming ball set is such that $U \in \mathcal{B}_m(X)$ iff $X \in \mathcal{B}_m(U)$, or more concisely we can write $I(U \in \mathcal{B}_m(X)) = I(X \in \mathcal{B}_m(U))$. Furthermore, the indicator function $I(U \in \mathcal{B}_m(X))$ factorizes as follows,

$$I(U \in \mathcal{B}_m(X)) = \prod_{i=1}^{N} I(\mathrm{h}(\mathbf{u}_i, \mathbf{x}_i) \leq m). \tag{6}$$

We wish now to consider $U$ as an auxiliary variable generated given $X$ uniformly inside $\mathcal{B}_m(X)$, i.e. we define the conditional distribution

$$p(U|X) = \frac{1}{\mathcal{Z}} I(U \in \mathcal{B}_m(X)), \tag{7}$$

where crucially the normalizing constant $\mathcal{Z}$ simply reflects the volume of the ball and is independent from $X$. We can augment the initial joint model density from Eq. (2) with the auxiliary variables $U$ and express the augmented model

$$p(Y, X, U) = p(Y|X)p(X)p(U|X). \tag{8}$$

Based on this, we can apply Gibbs sampling in the augmented space and iteratively sample $U$ from the posterior conditional, which is just $p(U|X)$, and then sample $X$ given the remaining variables. Sampling $p(U|X)$ is trivial as it requires to independently draw each $\mathbf{u}_i$, with $i = 1, \ldots, N$, from the uniform distribution proportional to $I(\mathrm{h}(\mathbf{u}_i, \mathbf{x}_i) \leq m)$, i.e. randomly select a $\mathbf{u}_i$ within Hamming distance at most $m$ from $\mathbf{x}_i$. Then, sampling $X$ is carried out by simulating from the following posterior conditional distribution

$$p(X|Y, U) \propto p(Y|X)p(X)p(U|X) \propto \left( \prod_{i=1}^{N} p(\mathbf{y}_i|\mathbf{x}_i) I(\mathrm{h}(\mathbf{x}_i, \mathbf{u}_i) \leq m) \right) p(X), \tag{9}$$

where we used Eq. (6). Exact sampling from this distribution can be done using the FF-BS algorithm in $O(M^2 N)$ time where $M$ is the size of each location-specific Hamming ball given in (4).

The intuition behind the above algorithm is the following. Sampling $p(U|X)$ given the current state $X$ can be thought of as an *exploration* step where $X$ is randomly perturbed to produce an auxiliary matrix $U$. We can imagine this as moving the Hamming ball that initially is centered at $X$ to a new location centered at $U$. Subsequently, we take a slice of the model by considering only the binary matrices that exist inside this new Hamming ball, centered at $U$, and draw an new state for $X$ by performing exact sampling in this sliced part of the model. Exact sampling is possible using the FF-BS recursion and it has an user-controllable time complexity that depends on the volume of the Hamming ball. An illustrative example of how the algorithm operates is given in Figure 2(b).

To be ergodic the above sampling scheme (under standard conditions) the auxiliary variable $U$ must be allowed to move away from the current $X^{(t)}$ (the value of $X$ at the $t$-th iteration) which implies that the radius $m$ must be strictly larger than zero. Furthermore, the maximum distance a new $X^{(t+1)}$ can travel away from the current $X^{(t)}$ in a single iteration is $2mN$ bits (assuming $m \leq K/2$). This is because resampling a $U$ given the current $X^{(t)}$ can select a $U$ that differs at most $mN$ bits from $X^{(t)}$, while subsequently sampling $X^{(t+1)}$ given $U$ further adds at most other $mN$ bits.

### 3.2 Extensions

So far we have defined Hamming ball sampling assuming binary factor chains in the FHMM. It is possible to generalize the whole approach to deal with factor chains that can take values in general finite discrete state spaces. Suppose that each hidden variable takes $P$ values so that the matrix $X \in \{1, \ldots, P\}^{K \times N}$. Exactly as in the binary case, the Hamming distance between the auxiliary vector $\mathbf{u}_i \in \{1, \ldots, P\}^K$ and the corresponding $i$-th column $\mathbf{x}_i$ of $X$ is the number of elements these two vectors disagree. Based on this we can define the $i$-th location Hamming ball of radius $m$ as the set of all $\mathbf{u}_i$s satisfying Eq. (3) which has cardinality

$$M = \sum_{j=0}^{m} (P-1)^j \binom{K}{j}. \tag{10}$$

This, for $m = 1$ is equal $(P-1)K+1$, for $m = 2$ it is equal to $(P-1)^2 \frac{K(K-1)}{2} + (P-1)K+1$ and so forth. Notice that for the binary case, where $P = 2$, all these expressions reduce to the ones from Section 3.1. Then, the sampling scheme from the previous section can be applied unchanged where in one step we sample $U$ given the current $X$ and in the second step we sample $X$ given $U$ using the FF-BS recursion.

Another direction of extending the method is to vary the structure of the uniform distribution $p(U|X)$ which essentially determines the exploration area around the current value of $X$. We can even add randomness in the structure of this distribution by further expanding the joint density in Eq. (8) with random variables that determine this structure. For instance, we can consider a distribution $p(m)$ over the radius $m$ that covers a range of possible values and then sample iteratively $(U, m)$ from $p(U|X,m)p(m)$ and $X$ from $p(X|Y,U,m) \propto p(Y|X)p(X)p(U|X,m)$. This scheme remains valid since essentially it is Gibbs sampling in an augmented probability model where we added the auxiliary variables $(U, m)$. In practical implementation, such a scheme would place high prior probability on small values of $m$ where sampling iterations would be fast to compute and enable efficient exploration of local structure but, with non-zero probabilities on larger values on $m$, the sampler could still periodically consider larger portions of the model space that would allow more significant changes to the configuration of $X$.

More generally, we can determine the structure of $p(U|X)$ through a set of radius constraints $\mathbf{m} = (m_1, \dots, m_Q)$ and base our sampling on the augmented density

$$p(Y, X, U, \mathbf{m}) = p(Y|X)p(X)p(U|X,\mathbf{m})p(\mathbf{m}). \tag{11}$$

For instance, we can choose $\mathbf{m} = (m_1, \dots, m_N)$ and consider $m_i$ as determining the radius of the $i$-location Hamming ball (for the column $\mathbf{x}_i$) so that the corresponding uniform distribution over $\mathbf{u}_i$ becomes $p(\mathbf{u}_i|\mathbf{x}_i, m_i) \propto I(\mathbf{h}(\mathbf{u}_i, \mathbf{x}_i) \le m_i)$. This could allow for asymmetric local moves where in some part of the hidden sequence (where $m_i$s are large) we allow for greater exploration compared to others where the exploration can be more constrained. This could lead to more efficient variations of the Hamming Ball sampler where the vector $\mathbf{m}$ could be automatically tuned during sampling to focus computational effort in regions of the sequence where there is most uncertainty in the underlying latent structure of $X$.

In a different direction, we could introduce the constraints $\mathbf{m} = (m_1, \dots, m_K)$ associated with the rows of $X$ instead of the columns. This can lead to obtain regular Gibbs sampling as a special case. In particular, if $p(\mathbf{m})$ is chosen so that in a random draw we pick a single $k$ such that $m_k = N$ and the rest $m_{k'} = 0$, then we essentially freeze all rows of $X$ apart from the $k$-th row[1] and thus allowing the subsequent step of sampling $X$ to reduce to exact sampling the $k$-th row of $X$ using the FF-BS recursion. Under this perspective, block Gibbs sampling for FHMMs can be seen as a special case of Hamming ball sampling.

Finally, there maybe utility in developing other proposals for sampling $U$ based on distributions other than the uniform approach used here. For example, a local exponentially weighted proposal of the form $p(U|X) \propto \prod_{i=1}^{N} \exp(-\lambda \mathrm{h}(\mathbf{u}_i, \mathbf{x}_i)) I(\mathrm{h}(\mathbf{u}_i, \mathbf{x}_i) \le m)$, would keep the centre of the proposed Hamming ball closer to its current location enabling more efficient exploration of local configurations. However, in developing alternative proposals, it is crucial that the normalizing constant of $p(U|X)$ is computed efficiently so that the overall time complexity remains $O(M^2 N)$.

## 4 Experiments

To demonstrate Hamming ball (HB) sampling we consider an additive FHMM as the one used in [6] and popularized recently for energy disaggregation applications [7, 10, 11]. In this model, each $k$-th factor chain interacts with the data through an associated mean vector $\mathbf{w}_k \in \mathbb{R}^D$ so that each observed output $\mathbf{y}_i$ is taken to be a noisy version of the sum of all factor vectors activated at time $i$:

$$\mathbf{y}_i = \mathbf{w}_0 + \sum_{k=1}^{K} \mathbf{w}_k x_{k,i} + \boldsymbol{\eta}_i, \tag{12}$$

where $\mathbf{w}_0$ is an extra bias term while $\boldsymbol{\eta}_i$ is white noise that typically follows a Gaussian: $\boldsymbol{\eta}_i \sim \mathcal{N}(\mathbf{0}, \sigma^2 I)$. Using this model we demonstrate the proposed method using an artificial dataset in Section 4.1 and a real dataset [11] in energy disaggregation in Section 4.2. In all examples, we compare HB with block Gibbs (BG) sampling.

## 4.1 Simulated dataset

Here, we wish to investigate the ability of HB and BG sampling schemes to efficient escape from local modes of the posterior distribution. We consider an artificial data sequence of length $N = 200$ generated as follows. We simulated $K = 5$ factor chains (with $v_k = 0.5$, $\rho_k = 0.05$, $k = 1, \ldots, 5$) which subsequently generated observations in the 25-dimensional space according to the additive FHMM from Eq. (12) assuming Gaussian noise with variance $\sigma^2 = 0.05$. The associated factor vector where selected to be $\mathbf{w}_k = w_k * \text{Mask}_k$ where $w_k = 0.8 + 0.05 * (k - 1), k = 1, \ldots, 5$ and $\text{Mask}_k$ denotes a 25-dimensional binary vector or a mask. All binary masks are displayed as $5 \times 5$ binary images in Figure 1(a) in the supplementary file together with few examples of generated data points. Finally, the bias term $\mathbf{w}_0$ was set to zero.

We assume that the ground-truth model parameters $\boldsymbol{\theta} = (\{v_k, \rho_k, \mathbf{w}_k, \}_{k=1}^K, \mathbf{w}_0, \sigma^2)$ that generated the data are known and our objective is to do posterior inference over the latent factors $X \in \{0, 1\}^{5 \times 200}$, i.e. to draw samples from the conditional posterior distribution $p(X|Y, \boldsymbol{\theta})$. Since the data have been produced with small noise variance, this exact posterior is highly picked with most all the probability mass concentrated on the single configuration $X_{\text{true}}$ that generated the data. So the question is whether BG and HB schemes will able to discover the "unknown" $X_{\text{true}}$ from a random initialization. We tested three block Gibbs sampling schemes: BG1, BG2 and BG3 that jointly sample blocks of rows of size one, two or three respectively. For each algorithm a full iteration is chosen to be a complete pass over all possible combinations of rows so that the time complexity per iteration for BG1 is $O(20N)$, for BG2 is $O(160N)$ and for BG3 is $O(640N)$. Regarding HB sampling we considered three schemes: HB1, HB2 and HB3 with radius $m = 1, 2$ and 3 respectively. The time complexities for these HB algorithms were $O(36N)$, $O(256N)$ and $O(676N)$. Notice that an exact sample from the posterior distribution can be drawn in $O(1024N)$ time.

We run all algorithms assuming the same random initialization $X^{(0)}$ so that each bit was chosen from the uniform distribution. Figure 3(a) shows the evolution of the error of misclassified bits in $X$, i.e. the number of bits the state $X^{(t)}$ disagrees with the ground-truth $X_{\text{true}}$. Clearly, HB2 and HB3 discover quickly the optimal solution with HB3 being slightly faster. HB1 is unable to discover the ground-truth but it outperforms BG1 and BG2. All the block Gibbs sampling schemes, including the most expensive BG3 one, failed to reach $X_{\text{true}}$.

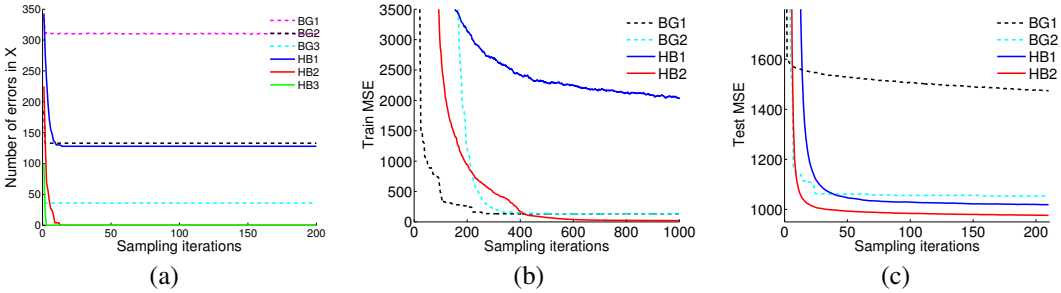

Figure 3: The panel in (a) shows the sampling evolution of the Hamming distance between $X_{\text{true}}$ and $X^{(t)}$ for the three block Gibbs samplers (dashed lines) and the HB schemes (solid lines). The panel in (b) shows the evolution of the MSE during the MCMC training phase for the REDD dataset. The two Gibbs samplers are shown with dashed lines while the two HB algorithms with solid lines. Similarly to (b), the plot in (c) displays the evolution of MSEs for the prediction phase in the REDD example where we only simulate the factors $X$.

## 4.2 Energy disaggregation

Here, we consider a real-world example from the field of energy disaggregation where the objective is to determine the component devices from an aggregated electricity signal. This technology is use-

ful because having a decomposition, into components for each device, of the total electricity usage in a household or building can be very informative to consumers and increase awareness of energy consumption which subsequently can lead to possibly energy savings. For full details regarding the energy disaggregation application see [7, 10, 11]. Next we consider a publicly available data set[2], called the Reference Energy Disaggregation Data Set (REDD) [11], to test the HB and BG sampling algorithms. The REDD data set contains several types of home electricity data for many different houses recorded during several weeks. Next, we will consider the main signal power of `house_1` for seven days which is a temporal signal of length $604, 800$ since power was recorded every second. We further downsampled this signal to every 9 seconds to obtain a sequence of $67, 200$ size in which we applied the FHMM described below.

Energy disaggregation can be naturally tackled by an additive FHMM framework, as realized in [10, 11], where an observed total electricity power $y_i$ at time instant $i$ is the sum of individual powers for all devices that are "on" at that time. Therefore, the observation model from Eq. (12) can be used to model this situation with the constraint that each device contribution $w_k$ (which is a scalar) is restricted to be non-negative. We assume an FHMM with $K = 10$ factors and we follow a Bayesian framework where each $w_k$ is parametrized by the exponential transformation, i.e. $w_k = e^{\widetilde{w}_k}$, and a vague zero-mean Gaussian prior is assigned on $\widetilde{w}_k$. To learn these factors we apply unsupervised learning using as training data the first day of recorded data. This involves applying an Metropolis-within-Gibbs type of MCMC algorithm that iterates between the following three steps: i) sampling $X$, ii) sampling each $\widetilde{w}_k$ individually using its own Gaussian proposal distribution and accepting or rejecting based on the M-H step and iii) sampling the noise variance $\sigma^2$ based on its conjugate Gamma posterior distribution. Notice that the step ii) involves adapting the variance of the Gaussian proposal to achieve an acceptance ratio between 20 and 40 percent following standard ideas from adaptive MCMC. For the first step we consider one of the following four algorithms: BG1, BG2, HB1 and HB2 defined in the previous section. Once the FHMM has been trained then we would like to do predictions and infer the posterior distribution over the hidden factors for a test sequence, that will consist of the remaining six days, according to

$$p(X_*|Y_*, Y) = \int p(X_*|Y_*, W, \sigma^2)p(W, \sigma^2|Y)dWd\sigma^2 \approx \frac{1}{T}\sum_{t=1}^{T} p(X_*|Y_*, W^{(t)}, (\sigma^2)^{(t)}), \quad (13)$$

where $Y_*$ denotes the test observations and $X_*$ the corresponding hidden sequence we wish to infer[3]. This computation requires to be able to simulate from $p(X_*|Y_*, W, \sigma^2)$ for a given fixed setting for the parameters $(W, \sigma^2)$. Such prediction step will tell us which factors are "on" at each time. Such factors could directly correspond to devices in the household, such as Electronics, Lighting, Refrigerator etc, however since our learning approach is purely unsupervised we will not attempt to establish correspondences between the inferred factors and the household appliances and, instead, we will focus on comparing the ability of the sampling algorithms to escape from local modes of the posterior distribution. To quantify such ability we will consider the mean squared error (MSE) between the model mean predictions and the actual data. Clearly, MSE for the test data can measure how well the model predicts the unseen electricity powers, while MSE at the training phase can indicate how well the chain mixes and reaches areas with high probability mass (where training data are reconstructed with small error). Figure 3(b) shows the evolution of MSE through the sampling iterations for the four MCMC algorithms used for training. Figure 3(c) shows the corresponding curves for the prediction phase, i.e. when sampling from $p(X_*|Y_*, W, \sigma^2)$ given a representative sample from the posterior $p(W, \sigma^2|Y)$. All four MSE curves in Figure 3(c) are produced by assuming the same setting for $(W, \sigma^2)$ so that any difference observed between the algorithms depends solely on the ability to sample from $p(X_*|Y_*, W, \sigma^2)$. Finally, Figure 4 shows illustrative plots on how we fit the data for all seven days (first row) and how we predict the test data on the second day (second row) together with corresponding inferred factors for the six most dominant hidden states (having the largest inferred $w_k$ values). The plots in Figure 4 were produced based on the HB2 output.

Some conclusions we can draw are the following. Firstly, Figure 3(c) clearly indicate that both HB algorithms for the prediction phase, where the factor weights $w_k$ are fixed and given, are much better than block Gibbs samplers in escaping from local modes and discovering hidden state configurations

that explain more efficiently the data. Moreover, HB2 is clearly better than HB1, as expected, since it considers larger global moves. When we are jointly sampling weights $w_k$ and their interacting latent binary states (as done in the training MCMC phase), then, as Figure 3(b) shows, block Gibbs samplers can move faster towards fitting the data and exploring local modes while HB schemes are slower in terms of that. Nevertheless, the HB2 algorithm eventually reaches an area with smaller MSE error than the block Gibbs samplers.

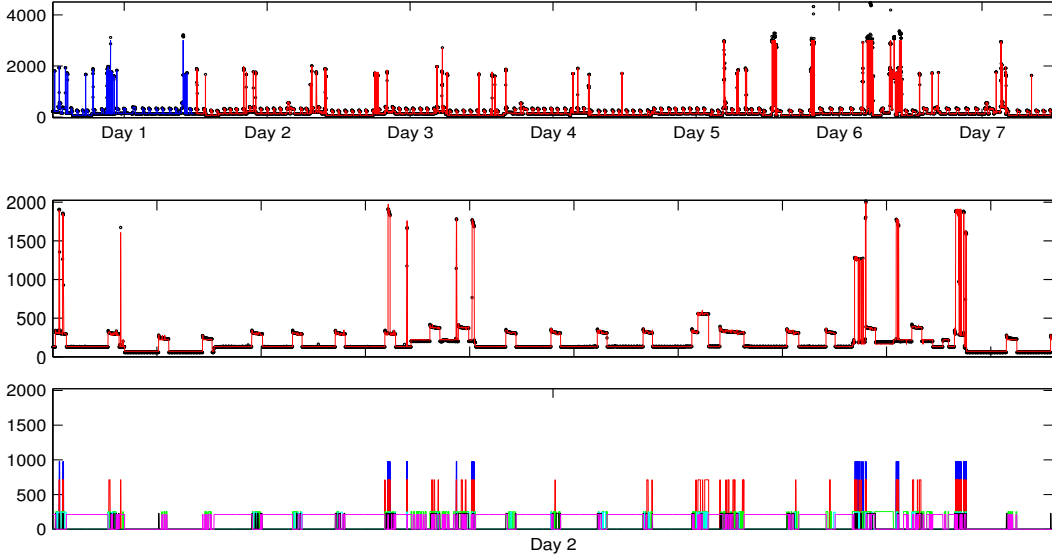

Figure 4: First row shows the data for all seven days together with the model predictions (the blue solid line corresponds to the training part and the red line to the test part). Second row zooms in the predictions for the second day, while the third row shows the corresponding activations of the six most dominant factors (displayed with different colors). All these results are based on the HB2 output.

## 5    Discussion

Exact sampling using FF-BS over the entire model space for the FHMM is intractable. Alternative solutions based on conditional updating approaches that use locally asymmetric moves will lead to poor mixing due to the sampler becoming trapped in local modes. We have shown that the Hamming ball sampler gives a relative improvement over conditional approaches through the use of locally symmetric moves that permits joint updating of hidden chains and improves mixing.

Whilst we have presented the Hamming ball sampler applied to the factorial hidden Markov model, it is applicable to any statistical model where the observed data vector $\mathbf{y}_i$ depends only on the $i$-th column of a binary latent variable matrix $X$ and observed data $Y$ and hence the joint density can be factored as $p(X, Y) \propto p(X) \prod_{i=1}^{N} p(\mathbf{y}_i | \mathbf{x}_i)$. Examples include the spike and slab variable selection models in Bayesian linear regression [12] and multiple membership models including Bayesian nonparametric models that utilize the Indian buffet process [13, 14]. While, in standard versions of these models, the columns of $X$ are independent and posterior inference is trivially parallelizable, the utility of the Hamming ball sampler arises where $K$ is large and sampling individual columns of $X$ is itself computationally very demanding. Other suitable models that might be applicable include more complex dependence structures that involve coupling between Markov chains and undirected dependencies.

#### Acknowledgments

We thank the reviewers for insightful comments. MKT greatly acknowledges support from "Research Funding at AUEB for Excellence and Extroversion, Action 1: 2012-2014". CY acknowledges the support of a UK Medical Research Council New Investigator Research Grant (Ref No. MR/L001411/1). CY is also affiliated with the Department of Statistics, University of Oxford.

## Footnotes

[1]In particular, for the rows $k' \ne k$ the corresponding uniform distribution over $u_{k',i}$s collapses to a point delta mass centred at the previous states $x_{k',i}$s.

[2]Available from `http://redd.csail.mit.edu/`.

[3]Notice that we have also assumed that the training and test sequences are conditionally independent given the model parameters $(W, \sigma^2)$.

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
