[Supplementary Material]

# Supplementary material for: Hamming Ball Auxiliary Sampling for Factorial Hidden Markov Models

## A    Extra plots from the experiment on the simulated dataset

The panel in Figure 1(a) shows the binary masks used to generate the simulated dataset in Section 4.1 (plots with labels $\mathbf{w}_1, \ldots, \mathbf{w}_5$) together with five data examples (remaining plots). Notice that the binary string below each example indicates the features that are present. The panel in (b) shows the true hidden sequence $X_{true}$ that generated the observations. The panel in (c) shows the three inferred hidden sequences (each considered to be the final value of $X$ after applying a large number of MCMC iterations) for the three block Gibbs samplers: the first plot from the top corresponds to BG1, the second to BG2 and the last to BG3. Notice that all these inferred sequences do not match the ground-truth shown in (b). Finally, the panel in (d) shows the corresponding inferred hidden sequences found by the HB algorithms: the first plot from the top corresponds to HB1, the second to HB2 and the last to HB3.

Figure 1: See main text for explanation.