[Reviews · NeurIPS 2014]

Submitted by Assigned_Reviewer_29

This paper presents a new Gibbs sampler algorithm for FHMMs. The idea is to add an auxillary variable, U, to the state of the Gibbs sampler. The value of U restricts the set of possible values that the hidden state X can take at the next step of the Gibbs sampler. As the number of possible values for X_i is small for each time point i, we can update X given U (and the data) using FFBS.

I think this is an original and clever approach to an important class of problems. The paper is written very clearly, and an empirical evaluation of the algorithm is done thoroughly. More importantly the intuition behind why (and when) this method would work well is presented. There are obvious extensions to this general idea -- which suggests the overall impact of the paper could be high.

E.g. one extension that may be possible is to define the set of values X_i could take given U in a way that depended on y_i (i.e. used likelihood information to so that you run FFBS allowing for the most likely states of X_i at each time-step).

I noticed a couple of typos

p.3 "four bits" -> "for four bits"
p.3 could you define Hamming distance?
p.4 "U as auxiliary" -> "U as an auxiliary"
Summary: A novel approach to MCMC for FHMM models: clearly presented and with a lot of scope to influence development of algorithms for this class of model.

Submitted by Assigned_Reviewer_32

This paper propose a new sampling algorithm for the state posterior distribution in a multi-chain model.

The original contribution of the paper relies on the application of slice sampling technique to the context of multi-chain hidden Markov models.

My major concerns are the following:
\begin{enumerate}
\item I wonder if the method can be extended to FHMM with more general transition matrix than the one given in eq. (1).
\item The illustrative example presented in Section 2 (see panel a) in Figure 2), shows that a pairwise blocked Gibbs cannot go in one step from the initial state multi-chain configuration to the true one. The authors claim that this is an example where the standard Gibbs chain is trapped by these type of local modes. I disagree with this conclusion. I would conclude instead that the example show that the mixing of a standard Gibbs chain can be slower than the mixing of the proposed Hamming ball sampler.
\item It has been shown that the mixing of the basic FFBS sampler for HMM can be improved by applying permutation sampler (Fr\"uwirth-Schnatter (2001)) or antithetic sampling (Billio et al. (2014)). I wonder if the performance of both the GB and the HB samplers can be further improved by combining them with these sampling strategies. Further simulation results (or at least a discussion) will be appreciated.
\end{enumerate}

Minor remarks:
\begin{enumerate}
\item line 152, provide a formal definition (or reference) for the Hamming distance.
\item line 168, The authors claim that "This Hamming ball, centered at
$X$, is simply the union of all $i$-th location Hamming balls of radius $m$", but then from the decomposition given in equation (6) it looks like it is an intersection.
\item line 171, is $I(\cdot)$ the indicator (or characteristic) function? I cannot understand why the authors define it as the identity function.
\item Bibliography needs a careful revision. E.g.: line 434, "Markov" instead of "markov"; line 452, "Monte Carlo" instead of "monte carlo"; line 467, "Indian" instead of "indian".
\end{enumerate}

References:
\begin{itemize}
\item Fr\"uhwirth-Schnatter, S., Markov Chain Monte Carlo Estimation of Classical and Dynamic Switching and Mixture Models, Journal of the American Statistical Association, Vol. 96, No. 453 (Mar., 2001),194-209
\item Billio, M., Casarin, R., Osuntuyi, A., Efficient Gibbs Sampling for Markov Switching GARCH Models, Comp. Stat. and Data Analysis, forthcoming.
\end{itemize}
Summary: The paper deals with a challenging issue and proposes an original sampler for the hidden states of a multi-chain HMM. The paper needs some minor revision, I would strongly suggest to accept it after revision.

Submitted by Assigned_Reviewer_41

This paper introduces a new sampling algorithm for Factorial Hidden Markov models. The standard algorithm for sampling the hidden chains in any HMM type of model is the forward filtering backward sampling algorithm. For many HMM like models this works very well; however for the FHMM, exact sampling from the joint distribution of all random variables in the hidden chain takes exponential time. One possible solution is to sample one or a few chains at a time keeping the other chains fixed. The problem with this approach is that this doesn't easily allow for multiple changes in multiple chains in one sampling pass.

This paper introduces a new algorithm called Hamming Ball Auxiliary Sampling. The idea of this sampling algorithm, is starting from the current sample X of all variables of the hidden chain, to introduce an auxiliary variable U which is a sample from the Hamming ball of a certain radius from the current sample where the Hamming ball is defined as any time step of all joint chain variables to be within Hamming distance m from the current sample. Next, the sample X is resampled conditional on U again by condition on being Hamming distance m away from U. As such, between two samples of X, at least 2*m bits can be flipped.

The paper is very well written and explains the algorithm well. There is a good empirical evaluation.

From experience, one problem with sampling from FHMM's is that if two parallel chains have very similar states, then it is likely that the parameters for those two chains will be very similar. Essentially, what should be one feature ends up being duplicated. This sampling algorithm does not make it more likely that this local maxima is escaped. I think it is still important to point out that that is still an issue (say around line 126).

l197: My only real complaint about the paper is that I'd like to see some algorithmic details on how this sampler can be efficiently implemented. I'm curious to know if the authors use an efficientl algorithm to sample from the Hamming ball? I.o.w. I'd add a section on implementation (and really I'd like to see some code published).

l215 - l230: I think this can easily be made an appendix or extension of the paper. It's a bit trivial and doesn't really make the paper that more impressive.

Section 4.1 As you pointed out, the different algorithms have different time complexities. I'd like to see Figure 3 scaled w.r.t. real complexity (rather than iterations which gives the HB algorithms an unfair advantage).

Nice work!
Summary: I think this is an important paper that makes good progress in efficiently sampling from the FHMM.
Author Feedback
Author rebuttal: Thank you for the positive feedback.

In response to the major issues:

1) Our presentation refers to a particular transition matrix structure but there is no reliance on this. The method is general and applies to FHMMs with arbitrary transition matrix.

2) We agree with the reviewer that it was imprecise to conclude that the Gibbs Sampler can become trapped by a local mode and agree that his alternate explanation is more accurate. We will make this change.

3) We were unaware of the permutation and antithetic samplers and will consider these in the final version.

4) We will ensure that implementation details are added in the final version and code is placed on an online code repository if the paper accepted.